# Contemporary Research Progress on the Detection of Polycyclic Aromatic Hydrocarbons

**DOI:** 10.3390/ijerph19052790

**Published:** 2022-02-27

**Authors:** Yan Zhang, Limin Yuan, Shuli He, Huilin Tao, Wenlian Xie, Xinyu Zhang, Xiaolu Ren, Tao Jiang, Lihong Li, Zhiqiang Zhu

**Affiliations:** 1School of Chemistry and Environmental Science, Shangrao Normal University, Shangrao 334001, China; hsl411724@163.com (S.H.); 18050782049@189.cn (H.T.); zhangxinyu202108@126.com (X.Z.); renxiaolu718@126.com (X.R.); 2School of Chemistry and Chemical Engineering, Shanghai University of Engineering Science, Shanghai 200335, China; yuan15932943096@163.com (L.Y.); xie15256971770@163.com (W.X.); lihongli0707@yahoo.com (L.L.); 3School of Environmental and Energy, Jiangxi Modern Polytechnic College, Nanchang 330095, China; zhangyan2006031201@163.com; 4National and Provincial Union Engineering Research Center for the Veterinary Herbal Medicine Resources and Initiative, Hunan Agricultural University, Changsha 410128, China

**Keywords:** polycyclic aromatic hydrocarbons (PAHs), analytical methods, ambient mass spectrometry, environmental pollution, microwave plasma torch (MPT)

## Abstract

Polycyclic aromatic hydrocarbons (PAHs) are a class of the most common and widespread contaminants. The accumulation of PAHs has made a certain impact on the environment and is seriously threatening human health. Numerous general analytical methods suitable for PAHs were developed. With the development of economy, the environmental problems of PAHs in modern society are more extensive and prominent, and attract more attention from environmental scientists and analysts. Deeper understanding of the properties of PAHs depends on the advent of detection methods, which can also be more conducive to promoting the protection of the environment. Till now, more sensitive, more high-speed and more high-throughput analytical tools are being invented and have played important roles in the research of PAHs. In this short review article, we focused mainly on the contemporary analytical methods about PAHs. We started with a brief review on the hazards, migration, distribution and traditional analysis methods of PAHs in recent years, including liquid chromatography, gas chromatography, surface enhanced Raman spectroscopy and so on. We also presented the applications of the modern ambient mass spectrometry, especially microwave plasma torch mass spectrometry, in the detection of PAHs, as well as the far out novel results in our lab by using microwave plasma torch (MPT) mass spectrometry; for example, some new insights about Birch reduction, regular hydrogen addition and the robustness of molecular structure. These studies have demonstrated the versatility of MPT MS as a platform in the research of PAHs.

## 1. Introduction

The prevention and control of environmental pollution is a sustaining hot point of contemporary society. As a class of the most common and widespread contaminants, polycyclic aromatic hydrocarbons (PAHs) are ubiquitous in a variety of environmental contexts, mainly including aqueous, air and solid samples that include sediments, soils and wastewater sludge [1,2,3,4]. PAHs are a general class of hydrocarbon compounds formed by condensing two or more benzene rings or cyclopentadiene rings, and appear important in the universe and interstellar medium, in which it is generally believed that about 10–25% of carbon is in the form of large polycyclic aromatic molecules [5,6]. Due to their relative stable structure, generic characteristics of PAHs are a high melting point, a high boiling point, and a low vapor pressure, and they are very difficult to metabolize and degrade in natural environment media, which leads to the accumulation of some PAHs for decades or even longer. Additionally, owing to their essential structure of polyphenyl rings, PAHs are virulent to the human beings. The most prominent hazards of PAHs are carcinogenicity, teratogenicity and mutagenicity. At present, thousands of carcinogens have been found in the world. Among the thousands of carcinogens, PAHs and their derivatives account for more than 30%, ranking them as the first class [7,8]. Usually, the carcinogenicity and biodegradability of PAHs increase with the increase in number of benzene rings of PAHs [9]. According to epidemiological studies, workers exposed to PAHs for a long time are prone to cancer, especially skin cancer, blood cancer, bladder cancer, nasopharyngeal cancer, gastric cancer as well as lung cancer [10]. Some surveys show that the death rate of lung cancer increases by 5% when the concentration of benzopyrene increases by 0.1 μg per 100 m^3^ [9,11]. Because PAHs are difficult to metabolize and degrade, they have a significant impact on human health and the ecological environment. The pollution of PAHs to the environment has persistently aroused the interest of researchers from different fields. Since the 1970s, generations of researchers have made great efforts and carried out a series of studies on the distribution, sources and risks in environmental pollution of PAHs.

In general, the sources of PAHs can be divided into two primary categories, natural sources and anthropogenic sources. The natural sources mainly include volcanic eruptions, grassland and incomplete forest combustion, as well as the synthesis of some organisms, and the proportion of this category in origin of PAHs is relatively light [12,13,14]. So, as the main source of PAHs, anthropogenic sources mainly include the incomplete combustion of motor vehicle fuel emissions and coal-burning emissions, which are both closely related to human’s daily life. For example, PAHs in cigarettes play a quite fundamental role in lung cancer of human beings. Perera et al. [15] showed that the level of B(a) p-DNA of benzopyrene adducts in the fetus was significantly higher than that in the mother, and the degree of DNA damage induction in the mother and the fetus showed a certain difference, 10 times as much as that in the mother. Jedrychowski et al. [16] made massive efforts and found that the offspring of mothers exposed to PAHs through diet in the third trimester had significantly lower birth weight and shorter birth length. Refer to some reviews and the articles cited in them for more toxicology of PAHs [17,18,19,20,21].

Clearly, understanding the mechanism of toxicity of PAHs and the far-reaching damage to the environment depends strongly on the advent of detection methods and analytical technologies, which are undoubtedly still challenging and contain new potential breakthroughs, whether in the analytical chemistry field or in environmental science, because PAHs usually exist in the environment as mixtures and in trace amounts [1]. Since the beginning of the new century, analytical techniques have been revolutionized, especially mass spectrometry, which was thought of as the best one for PAHs [1]. Ambient mass spectrometry will provide promisingly in situ on-line and high-sensitivity detection techniques for environmental scientists. We hope to summarize these advances in this aspect in this short review. After starting with a brief review on the hazards, migration, distribution and traditional analysis methods of PAHs which are widely used in recent years, including liquid chromatography, gas chromatography and surface enhanced Raman spectroscopy and so on, we mainly want to present the applications of the modern ambient mass spectrometry in the detection of PAHs; in addition, we present the far out novel results in our lab by using microwave plasma torch (MPT) mass spectrometry, which exhibited the versatility of MPT MS as a platform in the research of PAHs. MPT ion source is a novel ambient ion generator and has multiple advantages, for example, simple construction of the device and easy operation, low power dissipation, relative high sensitivity, and suitable for many types of samples including metal elements and organic samples [22,23,24,25,26].

## 2. The Spatial Distribution and Migration of PAHs

The presence of PAHs in almost all environmental media (atmosphere, water, soil) is due to the fact that PAHs can migrate and transport in all parts of the environmental ecophene [1,2]. PAHs are strongly lipophilic, when PAHs remain in the soil and accumulate in organisms through the food chain, they eventually cause certain harm to human body. The paths, through which PAHs can enter into the human body, comprise the following three channels: diet, skin contact, and respiratory tract [27]. For example, PAHs in the air can also interfere with children’s respiratory and nervous systems. Some researchers have revealed that increased respiratory susceptibility in non-allergic children in the early stages of exposure to pyrene [28]. Prolonged exposure to PAHs is an important cause of lung cancer in humans. Thus, PAHs pose a great threat and restriction to human development.

The understanding of the spatial distributions and migration mechanisms of PAHs will be beneficial for the detection and controlling of PAHs. On one hand, the more slowly PAHs migrate and transport, the more easily they can accumulate for a long time. As a result, they pose more threat to the environment, which makes the treatment of PAHs much more difficult. On the other hand, if the migration rate of PAHs is fast, the pollution range would be more extensive, and the effect of PAHs produced on the environment would be still important, and cause a burden to the analysis and control of PAHs. In short, it is of great importance to explore and understand the distribution and migration of PAHs, which also presents a consequential challenge to the analysis techniques for PAHs.

### 2.1. The Distribution, Migration and Harm of PAHs in Atmosphere

As PAHs are a semi-volatile pollutant, the atmosphere is the main receptor and transmission and diffusion channel for the emissions of PAHs, as well as an important reservoir of PAHs. The atmosphere is the most important environmental medium in the migration and transformation of PAHs, and is also the most important medium of human exposure to PAHs pollution [29,30,31]. PAHs in the atmosphere mainly come from pollutant emissions of industrial areas such as coking plants and incomplete combustion of civil coal-fired boilers [32,33]. Yuan-ju Li et al. [34] recently discovered that a large amount of PAHs is also produced during catering processing. Clarifying the emission characteristics of PAHs and the distribution characteristics of particles can provide an important basis for analyzing the generation mechanism and source of atmospheric secondary organic aerosols (SOA).

PAHs in soil and water also enter the atmosphere through evaporation. Some PAHs in the atmosphere will degrade under ultraviolet light, and also easily generate phenolic compounds, which will further react with NOx to generate nitrifies. Both Phenols and nitrifiers are the two main components of atmospheric SOA, and they will destroy the normal reaction cycle of NOx and O_3_, leading to a further increase in atmospheric O_3_ concentration. Therefore, PAHs is one of the reasons leading to serious atmospheric secondary pollution and high ozone concentration [34]. Atmospheric PAHs can also enter soil and water through descending migration. These processes are affected by their particle sizes, physical and chemical shapes, meteorological conditions and soil composition. The distribution of PAHs is different in different regions. Levels of PAHs pollution are generally higher in inland areas than in coastal areas because daily human production and living activities are closer to inland areas. With the change of different seasons, the concentration level of PAHs in the environment also varies to a certain extent [7], showing a general trend of high concentration in winter and low concentration in summer. In floating dust, inhalable particles adhere to most PAHs, thus they cause hidden dangers to the health development of humans and animals [8].

### 2.2. The Distribution, Migration and Harm of PAHs in Water

It was reported that about 89% of lakes in China are moderately polluted by PAHs, and 49% are severely polluted by PAHs [35]. PAHs have not only caused changes in sediment chemical nature and species structure, but also produced a series of negative effects on the development and utilization of water resources. Atmosphere deposition, sewage discharge and rainwater scouring are almost all of the pathways for PAHs to migrate into the natural lake water, and PAHs tend to end up in water in the form of sediments, since PAHs are less soluble in water. However, these sediments can also affect the surrounding ecosystem through the food chain, inevitably causing damage to human health.

### 2.3. The Distribution, Migration and Harm of PAHs in Soil

Soil is another important carrier of PAHs. Among the sources of PAHs in soil, besides natural sources, many human activities and industrial production take up most, including incomplete combustion of industrial fuels, atmospheric sedimentation and industrial sewage. Earlier studies [36] have shown that the atmospheric deposition of PAHs is the most important in this part, probably accounting for more than 90%. The content of PAHs in soil has noteworthy regional and seasonal differences, similar to that in air and water, which deeply indicates that PAHs migrates and converts among air, water and soil, forming an ecological closed loop. Multi-media migration is one of the important characteristics of semi-volatile organic pollutants such as PAHs, which also brings some difficulties to the research of PAHs. It is necessary to develop a holistic study and comprehensive analysis of various data, as well as simulating their migration and transformation behavior in all environmental media at the same time. The development of these new compatible methods and technologies relies more on modern analysis techniques and statistical processing methods from big data analysis.

PAHs enter soil, causing damage to the soil environment and normal working performance. PAHs in soil also have much more complicated environmental behaviors in soil, including adsorption, degradation and migration. Each process involves the influence of physical, chemical or biological environmental processes and is extremely complicated. For example, the adsorption of PAHs in soil affected by the soil surface chemical force on PAHs, as well as electrostatic force and van der Waals attraction from relatively long distance. Therefore, PAHs adsorption in soil has two different stages [37,38,39], namely fast process and slow process. The fast process leads to the adsorption of PAHs on the soil hydrophobic surface, while the slow process involves the migration of PAHs to the deep and inaccessible part of the soil matrix, which is easily absorbed and enriched by some vegetation [40,41], thus affecting the agricultural safety production to a certain extent. Different adsorption extent depends strongly on the physical and chemical properties of PAHs and soil [42], such as polycyclic aromatic hydrocarbons, water solubility, soil particle size, soil organic carbon content, pH and temperature. Adsorption of PAHs in soil in turn affectstheir further actions in the environment, such asvolatility, photolysis, hydrolysis, and the important factors in the process of biological degradation and so on. PAHs in soil are then transmitted through the food chain, eventually causing irreversible damage to humans and animals [43].

## 3. Traditional Analytical Methods for PAHs

Because of the strong carcinogenicity of PAHs, they had attracted attention from across the world. Researchers have been working on the efficient detection and degradation approach for PAHs for several decades. The rapid and sensitive analytical methods for PAHs keep persistent interest of analysts. However, the current studies in environment toxicology and physiological toxicology research just rest mainly in the preliminary stage, most of which had focused on the invention of some technology to degrade PAHs, especially on the efficiency of degradation. As for how to degrade, how to regulate the degradation process and so on, the research is still lacking. It is doubtless that these will promote the development of analytical methods for PAHs.

At present, in the environmental science and analytical testing field of the instrumental analytical methods for PAHs mainly incorporates high performance liquid chromatography (HPLC), gas chromatography (GC), capillary electrophoresis (CE), surface enhanced Raman spectroscopy (SERS), and optical spectroscopy as well as other analytical methods, for example, nuclear magnetic resonance spectroscopy (1HNMR), electrochemical method, nanopore technology, molecularly imprinted technique and so on. More detection techniques and related works can refer to a review article [44] and references therein. Among all the mature analytical methods, HPLC and GC are the most common ones and are generally chosen as the standard methods.

In this section, we reviewed briefly some applications in the detection of PAHs by using these methods. Noting the word “traditional” in the caption of this section, the authors want to clarify that these methods are not unique mass spectrometry, especially ambient mass spectrometry, which we will review in the next section.

### 3.1. High Performance Liquid Chromatography (HPLC)

The samples containing PAHs are usually of complex matrixes and some pretreatment procedures for separating samples are required in normal analytical methods. HPLC is one of the most common methods for the qualitativeandquantitative determination of PAHs. Within this method, one selectable liquid is employed as the high-speed mobile phase to achieve the purpose of separation based on their differential affinity between a solid stationary phase and a liquid mobile phase. The kinetics of distribution of solutes between the stationary and the mobile phase is largely diffusion-controlled [45].

There are a lot of applications of HPLC in PAHs detection. In this method, as Junlin Wang et al. [46] established, samples were filtered and enriched by glass fiber filter paper, and then performed via HPLC coupled with ultraviolet detector (UV) and sequent tandem fluorescence detector (FLR) to determine PAHs membranes in natural and tap water samples. The linear ranges of 16 kinds of PAHs were 0.5 to 500 ng·mL^−1^ and the correlation coefficients were greater than 0.999, determining the limit of detection (LOD) and limits of quantification (LOQ) as 0.3 to 5.0 ng·L^−1^ and 1.2 to 20.0 ng·L^−1^, respectively. The recoveries of water samples ranged from 67.2% to 114.1% with the relative standard deviations ranging from 1.5–14.0% (n = 6). The method has a sample operation system and good selectivity. It has achieved good results in the detection of PAHs in tap water and source water. Then, the established method was used for the determination of 17 water samples, 8 kinds, 6 kinds and 7 kinds of PAHs were detected in source water, tap water and pipe net tap water, respectively. These results exhibit this established method is of utility for the determination of 16 PAHs in source water and tap water.

Wen-Wu Yang et al. [47] developed a method which can determinate simultaneously 15 kinds of PAHs in barbecued meat in whichthe toast meat samples were purchased from different stalls in Chongqing city by using HPLC equipped with a fluorescence detection (HPLC-FLD). Samples were ultrasonically extracted, and the extract was purified by molecularly imprinted solid-phase extraction (MISPE), separated by HPLC and quantified by external standard method. Their results showed that the concentration of 15 PAHs in the range from 1 to 50 ng·mL^−1^ was linear with the chromatographic peak area with the correlation coefficients R more than 0.9995. Average recoveries at spiking levels of 5.0, 10.0 and 25.0 μg·kg^−1^ ranged from 71.1% to 98.8% with relative standard deviations (RSDs) from 1.0% to 5.8%. The LOD (S/N = 3) varied from 0.33 to 3.30 μg·kg^−1^ and the LOQ (S/N = 10) from 1.0 to 10.0 μg·kg^−1^.

After the water sample was extracted, Xiu-Qin Wang et al. [48] built an online system with HPLC for PAHs, then analyzed the tested substances by HPLC. Under optimum conditions, the online analysis method provided good linearity (0.03–30 μg·L^−1^), low detection limits (0.01–0.10 μg·L^−1^) and high enrichment factors (77.6–678). This method was applied to determine target analytes in river water and water sample of coal ash, and the recoveries are in the range of 80.6–106.6 and 80.9–103.5%, respectively. The recoveries of target analytes in river water and coal ash water samples were obtained.

Although HPLC is widely used in detecting PAHs, it has still some deficiency to be improved and perfected. First, the application of HPLC strongly depends on the detectors. The detector with high sensitivity, low detection limit and universality plays an important role in the detection and analysis of PAHs. Therefore, developing a suitable high-efficiency detector may be of certain significance. Second, high-powered separation device requires to be further developed. Currently, HPLC combined with solid phase membrane extraction has still some disadvantages to ameliorate, such as complicated pretreatment and expensive instruments and reagents. More importantly, the current hybrid HPLC methods can only show the information about the ingredients and content of various samples. However, the relevance and tangle of different ingredients are yet hidden in raw data. Exploiting deeply the information about the conversion, transition and synergy between different components of PAHs will be interesting and potential for the in-depth study involving PAHs.

### 3.2. Gas Chromatography (GC)

Gas chromatography (GC) is one of the most versatile and ubiquitous separating techniques in the laboratory. This instrumental method has the advantages of high selectivity, high sensitivity and simple operating system, and has a good resolving effect on the measured substances. In fact, it is widely used for the determination of organic compounds. Although the complex mixtures of PAHs are difficult to resolve because of the high degree of overlap in compound vapor pressures and boiling points, in recent years, GC has still been used more and more widely to detect efficiently PAHs. For example, the separation of benzene (boiling point 80.1 °C) is extremely simple by gas chromatography, but it is virtually impossible by conventional distillation. GC has indeed performed an indispensable role in agricultural monitoring, environmental monitoring, medical research and food identification.

Currently, many works on PAHs are employed by using GC coupled with different detectors. A typical kind of detector is mass spectrometer. As a highly sensitive analytical technology, gas chromatography-mass spectrometry (GC-MS) is often used to detect volatile and semi-volatile organic pollutants. This method combines gas chromatography and mass spectrometry; therefore, qualitative and quantitative analysis can be carried out simultaneously, but it is a bit hard to separate the isomers in PAHs. In the next section, we will show that some ambient mass spectrometry combined with some a plasma technique may be promising in directly analyzing the part isomers of PAHs.

Ming-Shan Zhang et al. [49] tested a way using a new 10 m short column of GC combined with the gas chromatograph-mass spectrometer (GC/MS) to determine 16 PAHs, with 5 deuterium internal standard and 2 substitutes in soil. Since the chromatograph column is short, the whole analysis of these samples was completed within 10 min. The rapid chromatogram is similar to that of the conventional method, but the response is faster. The liner relationship of 16 PAHs on the rapid column is good in the range of 5.0–400 μg·L^−1^ with the correlation coefficient R ≥ 0.997, and the LOD is 0.04–0.38 μg·kg^−1^. This method not only improves the detection efficiency, but also saves on cost.

Nam Vu-Duc et al. [50] introduced a capillary gas chromatography coupled with electron impact ionization tandem mass spectrometry (GC-EI-MS/MS) for the analysis of 16 PAHs in some particulate matter samples with the PAHs concentration between PM 2.5 and PM 10. The samples were extracted by ultrasonic-assisted liquid extraction and cleaned up by an acidic silica gel solid phase extraction. The linearity range of all analyzed PAHs was from 5 to 2000 ng·mL^−1^ with the square correlation coefficients R^2^ ≥ 0.9990. Limit of detection (LOD) of PAHs in particulate matter sample was from 0.001 ng·m^−3^ (2-bromonaphthalene) to 0.276 ng·m^−3^ (fluorene). The recoveries of PAHs in international proficiency testing sample ranged from 79.3% (chrysene) to 109.8% (indeno 1, 2, 3-cd pyrene). Based on the tandem mass spectrometry, they also found the main distribution of the PAHs in particulate matter samples was two-ring and three-ring compounds.

Anna Maria Sulej-suchomska et al. [51] established a reliable and accurate analytical method also based on headspace solid-phase micro-extraction coupled with comprehensive two-dimensional gas chromatography with time-of-flight mass spectrometry (GC×GC/TOF-MS) for simultaneous determination of 16 PAHs in 19 kinds of airport runoff water. The recovery obtained by this method was 63–108%, and mostly fell within the acceptable range for the analytical procedures. In addition, the developed procedure exhibited satisfactory selectivity, accuracy and low LOD values (0.22 ng·L^−1^ for Benzo[k]fluoranthene—2.20 ng·L^−1^ for phenanthrene, respectively). This method can be used to track the environmental fate of PAHs and to assess the environmental impact of airports.

Carlos Manzano et al. [52] developed a way to improve the separation of complex PAH mixtures (including 97 different parent, alkyl-, nitro-, oxy-, thio-, chloro-, bromo-, and high molecular weight PAHs), using a two-dimensional GC-MS, that is GC×GC/TOF-MS, by maximizing the orthogonality of different GC column combinations and improving the separation of PAHs from the sample matrix interferences, including unresolved complex mixtures. They used four different combinations of non-polar, polar, liquid crystal, and nano-stationary phase columns, andeach column combination can be optimized. They also evaluated each column for orthogonality using a method based on conditional entropy that considers the quantitative peak distribution in the entire 2D space (for completed chromatography data). Based on the approach, they analyzed the atmospheric particulate sample of matter PM 2.5 from Beijing, China, and a soil sample and a sediment sample from overseas for complex mixtures of PAHs. The found that the highest chromatographic resolution, lowest synentropy, highest orthogonality, and lowest interference from UCM were achieved by using a 10 m × 0.15 mm × 0.10 μm LC-50 liquid crystal column in the first dimension and a 1.2 m × 0.10 mm × 0.10 μm NSP-35 nano-stationary phase column in the second dimension. They demonstrated that the use of this column combination in GC × GC/TOF-MS resulted in significantly shorter analysis times (176 min) for complex PAH mixtures compared to 1D GC/MS (257 min), as well as potentially reduced sample preparation time, while their results merely contained the information of ingredients and contents of PAHs, without further exploring the correlations and synergy between different components of PAHs, in other words, the 2D-dimensional data is still underexplored.

Rosimeire Resendedos Santos et al. [53] have analyzed PAHs and their nitrification and oxidation derivatives in coffee by GC-MS. The recovery of this method was 82.1–96.3%, and the linear relationship was good. The correlation coefficient was R^2^ > 0.980. This method is easy to operate and has obvious advantages in the analysis and detection of coffee samples.

Zhihui Wu et al. [54] collected three different slime samples from three different regions of Xinjiang. These samples were extracted, eluted, dried and dissolved to determine the PAHs by GC-MS. A good linear correlation coefficient is obtained by this method with the square correlation coefficient R^2^ larger than 0.998. The detection limit was between 0.26 μg·kg^−1^ and 2.38 μg·kg^−1^. The study found that the higher the boiling points, the more PAHs were enriched in the sludge. Therefore, it is hopefully promising in serving as a useful method for testing PAHs in oil sludge.

W. Jira et al. [55] developed a GC-MS method for the analysis of 15 PAHs, including accelerated solvent extraction and the highly automated clean-up steps gel permeation chromatography and solid-phase extraction. In their studies, the six methylchrysene isomers and the PAH compounds with a molecular weight of 302 Daltons in fat-containing foods attained a better chromatographic separation with VF-17ms GC column. They also demonstrated the reliability of the analytical method for edible oils by the results from a proficiency test.

Except using mass spectrometer as the GC detector, there are still many works employing other traditional detectors to detect PAHs. Razieh Zakerian and Soleiman Bahar [56] prepared a graphene coating through electrochemical exfoliation of pencil graphite and then used it as a fiber coating for headspace solid-phase micro-extraction of PAHs from water samples; this was performed by the GC analysis technique with a flame ionization detector, since the flame ionization detector (GC-FID) works according to the principle of ions released in the combustion of the sample species, if there are any organic compounds. Under optimum conditions, the LOD range of six PAHs, including naphthalene, ranged from 0.01 μg·L^−1^ to 0.09 μg·L^−1^, and the linear ranges extend from 0.05–50 μg·L^−1^. The repeatability of the extraction process and the fiber-to-fiber reproducibility were in the ranges of 4.3–0.2% and 7.3–9.8%, respectively.

### 3.3. Capillary Electrophoresis (CE)

Capillary electrophoresis (CE) [57,58], known as high performance capillary electrophoresis (HPCE), is a new type of liquid-phase separation technology with capillary as the separation channel and high voltage dc electric field as the driving force. Actually, in CE, there involves electrophoresis, chromatography and the cross section of the capillary tube, and allows analytical chemistry to move from the micro-liter level to the nano-liter level. CE has been applied to the analysis of various samples because of its rapid, higher plate number and higher separation efficiency, economical and small injection volume, and convenient operation. It is certain that CE has been also developed in the detection of PAHs; moreover, CE technique can often provide the helpful supplement of GC for some high boiling point PAHs [59]. In addition, there are some derived CE techniques, such as micellar electrokinetic capillary chromatography (MEKC) [60,61] and parking capillary chromatography (PCC) [62], and they have been both applied in the studies of PAHs. However, for the sensitivity in this field, there is still plenty of room for improvement, especially when uniting Uv–Vis spectrometry, mainly owing to the tiny capillary diametermaking the light path too short. Maybe this can be overcome by optimizing the light path, such as employing multi-pass and so on. The biggest deficiency for CE may be that the reproducibility is poor due to the electroosmotic flow changing with the composition of the sample. From this point of view, it still requires a lot of effort.

Xin Chen et al. [59] used capillary electrophoresis to detect six kinds of PAHs, including fluoranthene and benzanthracene. After determining the optimal experimental conditions, such as the concentration of surfactant SDS, the composition of microemulsion and the time of high conductivity buffer HCB, they can rapidly determine the measured substance within 27 min. This method was used for the determination of brand cosmetics, and the recovery rate was between 90.6% and 95.9%, with the relative standard deviation of 3.3% to 5.1%.

Ludivine Ferey et al. [63] optimized cyclodextrin-modified capillary electrophoresis with laser-induced fluorescence detection. The utilizing of a dual CD system, involving a mixture of one neutral CD and one anionic CD, enabled them to reach unique selectivity and achieved the best separation effect of 19 PAHs with efficiencies superior to 1.5 × 10^5^ in 15 min for the first time. Additionally, the detection of PAHs in edible oil and real vegetable oil can be carried out by using internal standard with low LOQs in mg·L^−1^. Importantly, the authors claimed that using umbelliferone as an internal standard with appropriate electrolyte and sample compositions, rinse sequences and sample vial material, they significantly improved the repeatability.

Amanda M. Stockton et al. [64] developed the Mars Organic Analyzer (MOA), a portable microchip CE instrument to analyze laboratory standards and real-world samples for PAHs. The LOD for the PAH components of the standard ranged from 2000 ppm to 6 ppb. This work established the viability of the MOA for detecting and analyzing PAHs in in situ planetary exploration.

### 3.4. Surface Enhanced Raman Spectroscopy (SERS)

Raman spectrum [65] belongs to molecular vibration spectrum reflecting the characteristic structure of molecules. However, the Raman scattering effect is an extremely weak process, and the scattering light intensity is only about 10^−10^ of the incident light intensity. Therefore, the Raman spectroscopy does work well with the surface-adsorbed species producing some kind of enhancement effect, i.e., surface-enhanced Raman spectroscopy (SERS) [66,67]. SERS not only has the advantage of Raman spectroscopy, but also significantly improves the signal intensity of molecules on the basis of Raman spectroscopy. After more than 40 years of development, SERS has been widely used in various fields by virtue of its advantages such as convenience, low detection limit and high sensitivity. Utilizing the adsorption of PAHs on the surface of rough noble metal substrate can markedly enhance the Raman signal, thus, SERS is suitable for the detection of PAHs.

Shan Wang et al. [68] had successfully used gold nanofilms as SERS substrate to establish a method for the determination of three kinds of PAHs, such as pyrene in water samples using SERS. The rapid detection and analysis of the sample was completed within 5 min. The LOD values for naphthalene and pyrene are 10 ng·L^−1^ and for m-triphenyl is 50 ng·L^−1^, respectively. This detection method has a good prospect in food safety and environmental monitoring.

Adopting a hyphenated technique combining SERS with surface micro-extraction byusing methanol and 1-propanethiol-modified silver nanoparticles, Min Zhang et al. [69] realized the in situ analysis of PAHs on food contact materials. In the study, the characteristic vibration of the C–C bond of parathyroid hormone at 1030 cm^−1^ was used as the internal standard for quantitative determination in this method, giving high uniform 1-propanethiol-modified silver nanoparticles. In the concentration range of 0.789–158 ng·cm^2^, the standardized SERS intensity showed a good linear relationship with fluoranthene concentration, the LOD was 0.27 ng·cm^2^. This detection method can realize the rapid screening of PAHs mixtures, greatly improving the analysis efficiency.

Yong-Hyok Kronfeldt et al. [70] prepared the sol–gel matrix embedding Ag nanoparticles functionalized with 25, 27-dimercaptoacetic acid-26, 28-dihydroxy-4- tert-butylcalix (4) arene (DMCX) via thermal reduction method and applied it in the detection of PAHs in seawater. DMCX forming the monolayer on the silver nanoparticle surface contributes to the surface-enhanced Raman scattering (SERS) activity due to the aggregation of silver nanoparticles and the pre-concentration of PAH molecules within the zone of electromagnetic enhancement. A calibration procedure reveals that this type of SERS substrate has a limit of detection of 3 × 10^−10^ mol·L^−1^ for pyrene and 1.3 × 10^−8^ mol·L^−1^ for naphthalene in artificial seawater. The greatest feature of this method can be in situ and on-line. In 2020, Zheng-Dong Shen et al. [71] combined an on-chip thin-layer chromatography and SERS to identify successfully PAHs from cooking oil samples without sample pretreatment. In their experiments, the SERS LOD can reach at 1 ng per spot.

### 3.5. Optical Spectrometry

Optical spectrometry is a major category of commonly used instrumental methods covering almost all analytical and testing fields with the advantages of nondestructive and operation easily. Optical spectrometry has been used to detect PAHs for a long time, involving mainly fluorescence spectrometry, phosphorescence spectrometry, and spectrophotometer. Methodologically speaking, the former two pertain to emission spectrometry, similar to SERS, more sensitive, while the last one belongs to absorption spectrometry.

In general, fluorescence [72] is more likely related to those molecules with specific structures, such as planar or rigid electron-conjugated systems. PAHs are just the most representative molecules of these types of structures. Since spectral data provide the molecular “fingerprint” information, different PAHs have different spectral characteristics, the detection of PAHs can be realized simultaneously by fluorescence spectroscopy.

Li-Fang He et al. [73] established a fast analytical method for simultaneous identification of PAHs in water, in which a novel method of constant wavelength synchronous fluorescence was proposed to the simultaneous determination of different PAHs in a mixture of 14 components. The linear response of this method was in the range of 0.1–1000 ng·mL^−1^ (R ≥ 0. 9988), and the relative standard deviations (RSD) were in the range of 1.06% –1.67% (n = 6). The LODs were in the range of 0. 072–3.9 ng·mL^−1^.

Qi-Hong Cai et al. [74] established a new method for simultaneously determining five PAHs—fluorene, benzofluorene, pyrene, benzo (a) pyrene and perylene—in dairy products by constant-wavelength synchronous fluorescence spectrometry (CWSFS), without the need for previous chromatographic separation of the analyte solution. After ultrasonic extraction of the five PAHs from the dairy products using acetonitrile as the extraction solvent, the supernatants were filtered by 0.45 μm micro-porous filter membrane and concentrated to dry by a nitrogen dryer. They also chose the difference of wavelength (Δλ) 40 nm in CWSFS scanning to overcome the interference from the background matrix and between PAHs. The LODs of this method can arrive at 0.016 μg·L^−1^ for some PAH species.

Guo-Ying Wang et al. [75] pretreated an ultrasonic extraction method to achieve rapid quantitative analysis of PAHs in atmosphere particles by constant energy synchronous fluorescence spectroscopy. The fluorescence emission spectrum of PAHs can be simplified by using constant-energy synchronous excitation, and the Rayleigh scattering interference of the solution can be effectively solved. Quantitative analysis was performed at the optimal energy difference for the PAHs with LOD and LOQ of 0.0580–3.18 and 0.232 –12.7 ng·mL^−1^, respectively. The recoveries of the 15 PAHs in the blank and at certain concentrations ranged from 82.8% to 120.0%, and the relative standard deviations ranged from 0.51% to 5.87%.

However, PAHs are difficult to participate in chemiluminescence reactions, and usually the application of fluorescence analysis and luminescence analysis in the detection of PAHs is yet limited. Huan-Bo Wang et al. [76] established an excited-emission matrix fluorescence array for the detection of anthracene, pyrene, fluoranthene, phenanthrene and fluorene, but it was inevitably interfered by humic acid and falvic acid in the actual sample analysis. Jian-Xu Li et al. [77] used silver nanoparticles to modify the titanium dioxide nanotube electrode, and the established electroluminescence method can sensitively detect PAHs for more than four benzene rings, but this method is difficult to detect benzene, naphthalene and anthracene since they are not easily oxidized. Fluorescence analysis is also used by several research groups to detect PAHs.

In principle, fluorescence and phosphorescence [78,79,80,81] can be viewed as mutual duals because fluorescence originates from the transition of a singlet state to another singlet state, while phosphorescence comes from the transition of a singlet state to triplet state. Comparing with fluorescence signal, phosphorescence signal is generally much weaker, but the lasing time is relatively longer. Based on this characteristic, the phosphorescence can be applied in the detection of PAHs. A. Segura Carretero et al. [82] studied the detailed characterization of the microemulsion composition, which is necessary to make the phosphorimetry suitable as powerful analytical methodology for PHAs. They also established a method simultaneously determined five PAHs (acenaphthene, fluoranthene, pyrene, benz (a) anthracene and benzo (a) pyrene) by variable-angle synchronous scanning (VASS) microemulsion phosphorimetry at room temperature [83]. In order to obtain the optimum phosphorescence responses, a statistical model of central composite design type, was applied. Therein, the VASS technique employed enhanced the selectivity permitting the simultaneous determination after addition to road dust samples giving mean recoveries of 87.6% with a relative standard deviation of 3.0% at n = 5.

In 2020, Elham Mansouri et al. [84] critically and objectively reviewed ultraviolet-based methods especially in high-performance liquid chromatography-ultraviolet. They mainly discussed the high-performance liquid chromatography ultraviolet-diode array detector and ultraviolet-fluorescent detector and their applications in scientific studies to measure polycyclic aromatic hydrocarbons concentration. The review also gives useful and comprehensive information about valuable methods for future research on PAHs.

More traditional spectrophotometry also was used to analyze the PAHs. Additionally, by combining with chemometrics, spectrophotometer can also make a big difference. Li-Xin Luan et al. [85] used principal component regression analysis as well as ultraviolet spectrophotometer to analyze the PAHs content in white oil. The total PAHs content can be determined directly without considering the problem of spectral overlap. The recoveries of the method fall into 92.41–99.01%.

### 3.6. Other Analytical Methods

Except those methods mentioned above, there are still other ones applied in the research of PAHs, too. Here, we only give a simple comment. Electrochemistry is also a common analytical method [86,87]. Xiao-Fang Shen et al. [88] fabricated a sensor combining pre-concentration and in situ electrochemical determination based on electropolymerized poly(3-methylthiophene) (P3MT) to determine the concentration of 1-Hydroxypyrene, which is widely used to assess exposure to PAHs. Eric J. Moore’s group [89] optimized the detection technology of biosensor and used linear sweep voltammetry to analyze phenanthrene in environmental water samples with the LOD of 1.4 ppb. Yong-Nian Ni et al. [90] constructed a novel layer-by-layer electrochemical biosensor, DNA/hemin/nafion–grapheme/GCE, for the analysis of the benzo (a) pyrene PAH; the concentration in aqueous solution was determined by differential pulse voltammetry based on the linear relationship between metabolites and concentration of benzo (a) pyrene in the presence of hydrogen peroxide.

Nanopore technology [91] is another used in the detection of PAHs, especially benzo (a) pyrene. In this technology, macro-molecules flow under outside pressure through the nanopores made of insulating material to produce the blocking of the ion flow through the holes and lead simultaneously to the instantaneous change in the conductivity of pores, reflecting the current value by the size of the particles. The value of the change reflects the number of biological macromolecules. Rukshan T. Perera et al. [92] constructed an α-hemolysin (αHL) nanopore platform which can be used to detect the benzo(a) pyrene diol epoxide adducts to guanine in synthetic oligodeoxynucleotides by producing a unique multi-level current signature. This study presented opportunities for the monitoring, quantification, and sequencing of mutagenic compounds from cellular DNA samples.

Molecular imprinting technology (MIT) [93] is another important analytical method, which is used to entrap analytes of interest for the subsequent detection. Usually, it uses molecularly imprinted polymers to simulate the interaction between enzyme–substrate or antibody–antigen to recognize specifically printed molecules. Thus, MIT has also been used in the analysis of PAHs. Hao Li et al. [94] performed high-selective luminescence detection of trace PAHs based on the specificity of molecularly imprinted polymers and magnetic separation. Phenanthrene of double benzene can be determined low limited at 3.64 ng·mL^−1^ by the preparation of optomagnetic multifunctional molecularly imprinted polymers using polystyrene-methacrylic acid copolymer, hydrophobic Fe_3_O_4_ nanomaterial and luminescent LaVO_4_: Eu^3+^ nanomaterial. Additionally, the recovery obtained by adding phenanthrene into some milk was 97.1–101.9%.

## 4. Mass Spectrometry

The detection of PAHs in the environment mostly involves trace levels or ultra-trace levels and has to face diverse complex matrixes, in which a large number of different substrates interfere and coexist. Therefore, the matrix effect and the detection efficiency except those routine indexes must be considered in developing new applicative and powerful analytical tools. Mass spectrometry [95,96] is a measuring tool for directly determining the molecular weight of the analyte with high sensitivity, high precise and high speed. Owing to this fundamental nature, modern mass spectrometry is more and more widely used in various fields related to every aspect of natural science and daily social life. A typical early work was finished by R. Zenobi’s group [97], who employed an ionization technique called resonance enhanced multi-photon ionization (REMPI) coupled with a time-of-flight mass spectrometer to measure quantificationally some PAHs with three-six benzene rings in water covering the range of 2–125 ng·L^−1^. The recovery ranged from 75% to 90%. Xiao-Xiang Zhang et al. [98] built a self-regulating TOF combined with the vacuum ultraviolet photo ionization which can realize the online detection of some small molecular PAHs.

However, early applications of mass spectrometry in the studies of PAHs may depend on the tedious and fussy pretreatment procedure and the cumbersome equipment for the ionization of analytes. A real contemporary revolution in the mass spectrometry, especially in ion sources, was the development of ambient ionization techniques, called ambient mass spectrometry [99,100], which not only did extend simple sample-content testing to the in situ analysis even to surface imaging, but also enabled the ionization of samples in their native environment without sample pretreatment and brought the breakthrough in the application of MS to high-through analysis. After the emergence of the pioneered technique, desorption electrospray ionization (DESI) [101], more than dozens of direct-ionization techniques have emerged which can meet the requirements of being real-time, in situ, online, non-destructive, and in accordance with the new concept of no pollution and low energy consumption. Typical ambient ion source includes direct analysis in real time (DART) [102,103], surface desorption atmospheric pressure chemical ionization (DAPCI) [104,105], extractive electrospray ionization (EESI) [106,107,108], dielectric barrier discharge ionization (DBDI) [109], flowing atmospheric pressure afterglow (FAPA) [110], etc., all of which have obtained a series of achievements in the analysis of complex matrix samples and greatly expanded the range of analytical objects of mass spectrometry, from various fields including metabolomics [111,112,113], proteomics [114,115], forensic medicine [116,117], and quality monitoring [118].

Since the main function of ambient ion sources is to produce the analyte ions under atmospheric condition, i.e., translate the neutral analyte molecules into positive and negative ion forms. that are generating the plasma at ambient air. Among these ambient ion sources, there are a class of ion sources directly generating the visible plasma [119], including plasma-assisted desorption ionization (PADI) [120], low-temperature plasma (LTP) ionization [121], microplasma discharge ionization [122], and desorption corona beam ionization (DCBI) [123]. Among them, a remarkable high-frequency plasma source, microwave plasma torch (MPT), was invented initially by Jin’s Group [124], and was substantially promoted by Gary M. Hieftje’s group [125]. MPT easily generated a stable and visible flame-like plasma at atmospheric pressure and operated at commercial 2.45 GHz, by which the plasma operation was significantly improved [126]. MPT offered a much better analytical performance for the introduction of aqueous aerosols. At the incipient stage, MPT were mainly used as the emission light source for atomic emission spectrometry (AES), portable spectrometer [127,128], supercritical fluid chromatography (SFC) [129], or liquid chromatography (LC) [130]. However, MPT processes relatively high ionization efficiency, although less than 100%, and it also plays an important role in massspectrometry. In an early work, Gray M. Hieftje et al. built an MPT-TOF MS for the detection of halogenated hydrocarbons separated by a capillary gas chromatography [131]. After a long time, Ti-Qiang Zhang as well as Zhi-Qiang Zhu [22,23,132] studied the direct desorption/ionization approach of MPT on a linear ion trap (LTQ) mass spectrometer to analyze a series of small organic molecules, and they showed that MPT is a useful alternative ambient ion source. Zhi-Qiang Zhu’s group [24,25,26,133,134,135,136] has also completed a lot of work on the detection of trace metal in aqueous solutions by MPT source, coupled with a mass spectrometer (MPT-MS), reaching an in-depth understanding of the characteristics of MPT.

After that, Zhiqiang Zhu et al. turned to the studies on PAHs by MPT-MS. In this section, we mainly presented partially interesting results on benzene and one groupof isomers of PAHs, and provided new sights on Birch reduction, regular hydrogen addition and molecular robustness. We also demonstrated that the MPT mass spectrometry is promising in the quick detection of PAHs, especially in directly distinguishing some isomers of PAHs.

### 4.1. MPT Mass Spectra of Benzene

Strictly speaking, benzene does not belong to PAHs, since it is merely of single ring and is the fundamental unit of PAHs, and the knowledge on its MPT mass spectra will be beneficial for the deep understanding of PAHs.

As early as in 2009, Na Na and R. Graham Cooks et al. [137] found the Birch reduction in benzene, even toluene and naphthalene, in low-temperature plasma generated by a dielectric barrier discharge (DBDI) ion source. Under their experimental conditions, the product via Birch reduction is the major yields with about 71% productivity. Birch reduction usually refers to the reduction in aromatic rings to 1,4-cyclohexadiene via a specific hydrogen addition reaction in liquid phase in the presence of sodium and alcohol, that is they observed *m*/*z* 80 as the main peak in benzene mass spectra. In the previous studies, this particular kind of hydrogen addition was closely related to the aromaticity of the molecule, a special stability due to the annular conjugated system, in which the aromatic ring firstly attracts free electrons provided by the alkali metals and is mediated through liquid NH_3_ to generate energetic radicals [138]. However, the initial trigger of Birch reduction, the free electron, is quite easily available in surrounding plasma. Therefore, Na Na’s work is actually the first to find Birch reduction in the gas phase (more strictly, in the plasma phase) and really provides a new technical approach to study the hydrogen addition reactions involving PAHs. By the way, in their research, they also spent a great deal of effort verifying that the source of H for addition in their reaction system was from a silicon wafer, but not from benzene or ambient surrounding.

In MPT plasma, does Birch reduction still occur? In other words, is Birch reduction unique in LTP plasma? Firstly, it is worth emphasizing on that these two kinds of plasma, MPT and LTP, are not fully identical. The biggest one is the excitation temperature. The excitation temperature in MPT plasma is relatively high, several hundred Kelvin up to 2000 K [138], while the corresponding one in LTP is merely at room-temperature level [139], at which a person’s skin can withstand (that is why it was named low-temperature). Such a huge gap in temperature is responsible for the difference in the mechanisms and products in plasma.

Herein, Figure 1A shows the first order mass spectrum of benzene obtained by using MPT as the mass source, coupled with a linear ion trap mass spectrometer (MPT-LTQ MS) in positive mode. Comparing with Figure 1A, in MPT mass spectrum, there are plenty more signals related to benzene. The first set of peaks contains *m*/*z* 78, 79 and 80, in which the former two also disappeared in the result of Na Na et al. They are obviously assigned to benzene cation, [M]^+^, protonated benzene [M+H]^+^ and hydrogen addition benzene [M+2H]^+^, respectively. Here, M represents benzene. Noting that the intensity of the peak *m*/*z* 80 is very weak, it seems likely that regular hydrogen addition or Birch reduction is difficult in current conditions, i.e., in MPT plasma. However, there is a second set of peaks comprising of *m*/*z* 81, 82 and 83, wherein the signal of *m*/*z* 81 is remarkable while *m*/*z* 80 is very feeble, which is completely different from the result of Na Na et al. One reasonable explanation is that the peak of *m*/*z* 81 is assigned to the protonation of the product of benzene by hydrogen addition, [M+3H]^+^, and this protonation rate is so fast that it is the spilled away from the hydrogen addition, which causes the near disappearance of the *m*/*z* 80 ions. At present, it is not yet clear whether this hydrogen addition is just a regular reaction or Birch reduction. Just thinking from the viewpoint of the electron cloud configuration and π electron bonding, the product via the regular hydrogen addition is more likely to combine easily with the additional proton to form relatively stable p-π bond, which explains the formation of *m*/*z* 81 as well as the disappearance of *m*/*z* 80. These possible underlying mechanisms were all illustrated in the Figure 1 below. Of course, there are other alternative mechanisms, such as hydrogen addition occurring in the protonation of benzene, [M+H]^+^. More detail process validation and quantum chemistry simulation are working on it. Regardless of the reaction mechanism, Figure 1 illustrates the wide difference between the MPT and LTP mass spectra of benzene. Therefore, the former enriches the detection methods of benzene and paves the way to the research of PAHs by MPT mass spectrometry.

In Figure 1, the predominant signal is *m*/*z* 94 as well as the associated peaks of *m*/*z* 95 and 97, which are seemingly the parallel shift of *m*/*z* 78, 79 and 81, by pulsing an oxygen atom. This is another distinct feature in the MPT plasma comparing with that in LTP plasma. There are still some larger ions, that is *m*/*z* 110 and bigger ones. These bigger ions are unclear till now, meaning some new synthetic reactions are taking place in MPT plasma. Some possible assignments are shown in the inset. In brief, MPT mass spectrum of benzene can present more information and insight about the details of these reactions of PAHs in plasma.

Figure 1B shows a second-order mass spectrum of benzene (MPT-LTQ-MS^2^), the selected precursor *m*/*z* 79, [M+H]^+^. Additionally, the inset shows the deduced dissociative sequences in CID (collision-induced dissociation). As shown, there are several possible dissociative paths pointing to the finial main product, *m*/*z* 79, C_4_H_3_^+^.

### 4.2. MPT Mass Spectra of Fluoranthene and Pyrene

Usually, directly screening the two isomers is difficult and tedious in mass spectrometry. In PAHs, fluoranthene and pyrene are a pair isomer with multiple aromatic rings with relative molecular weights 202 Da. They are both potent cocarcinogens when applied together with benzo(a)pyrene on mouse skin [140]. These two isomers are both nonromantic since the number of π electrons is 16 and does not meet the famous Huckel’s rule; thus, there should be no Birch reduction occurring for these two PHAs, at least in MPT plasma. Figure 2 shows these experimental results by a MPT source coupled with a miniature time-of-flight mass spectrometer (Guangzhou Hexin Instrument Co., Ltd., see Figure 3 for the experimental device interface). Figure 3A is the MPT mass spectrum of fluoranthene, in which, the main peak, *m*/*z* 202 is the molecular ion of fluoranthene, and *m*/*z* 203 is the protonation of fluoranthene, [M+H]^+^. The peak possible representing Birch reduction or hydrogen addition, *m*/*z* 204, is almost invisible. While another peak related to hydrogen addition, *m*/*z* 205, maintains a certain intensity, which is likely to originate from the fast protonation of *m*/*z* 204 or the hydrogen addition of the ions of *m*/*z* 203, or both. This phenomenon is in accord with that of benzene. On the other hand, in Figure 3B, the MPT mass spectrum of pyreneexhibits a slightly different feature. The characteristic peaks, including *m*/*z* 202, 203, 205, are all similar to those of fluoranthene, but the peak of *m*/*z* 205 is dominated in the range of 150–250 Th, and there are some non-negligible ions of *m*/*z* 206 and 207. Another distinct feature is that there are more abundant fragmental signals.

The distinction of the two mass spectra can be easily explained and well consistent with the discussions above. The signal of *m*/*z* 202, 203, 205 accrue naturally, similar with that in benzene. The difference in abundance of these peaks between fluoranthene and pyrene may be due to their molecular robustness. From the viewpoint of molecular structure, though fluoranthene and pyrene are both planar molecules, pyrene’s molecular plane is more rigid, since its four benzene rings pack closely. There is hardly any bending vibration mode deviated from the plane (see Appendix A). By contrast, in fluoranthene, two benzene rings stack in turn and connect with the third benzene ring through two carbon–carbon single bonds. This structure is flexible, with several bending vibration modes deviated from the plane (see Appendix A). Some vibration modes and their corresponding frequencies are listed in Appendix A. It is visual that rigid molecules are prone to break and produce fragments or derivatives under collisions as long as the collision energy is strong enough. The flexible molecules are likely to survive in collisions at the same conditions. Therefore, it is not difficult to understand that the fragments in Figure 3A are relatively rare, while the fragments in Figure 3B are relatively plenty. Additionally, for the same reason, hydrogen addition is more likely to happen with more rigid molecules; therefore, pyrene can have multiple hydrogen addition to yield *m*/*z* 206 and 207. The disappearance of *m*/*z* 204 corresponding to the direct hydrogen addition attributes to the fast protonation to form stable p-π bond. Of course, detailed quantum chemistry computation is still needed to validate these explanations. Nevertheless, the MPT mass spectrometer can provide a facility to quickly screen some isomers of PAHs.

## 5. Prospects

In this short article, we review the contemporary analytical methods for PAHs. The detection of PAHs is undoubtedly important in modern social life and scientific research. There are also many ways to detect PAHs. Unfortunately, some of them are flawed in traditional analysis methods. For example, high performance liquid chromatography pretreatment is excessively complicated, and the detection sensitivity of capillary electrophoresis requires still to be further improved. There are some analytical methods which have not yet played their full role. For example, GC×GC-MS, in which the data is extended one-dimensional, is more suitable for exploring the implicit complicated relationships between multiple factors such as synergistic competition or much higher order relations. However, it does not exert a prominent effect and achieve a breakthrough case, although it has become an important equipment for field atmospheric monitoring [43,141]. In addition, it is still worth noting that there are still many novel powerful analytical approaches, such as micro-fluidics [142,143], chemometrics [144,145], and machine learning [146,147] and so on. These new approaches have been applied in numerous aspects of contemporary leading-edge research. These techniques or methods all have huge potential in complex matrix samples, while they can also be applied in the detection of PHAs. Limited to the author’s knowledge, these methods and technologies have rarely been used in the studies of PAHs, thus we do not mention them, regrettably. However, it is certain that they will shine in the research of PAHs in the near future.

Completely realizing real-time and online detection and analysis at least comprises direct sample injection and ionization technique under atmospheric pressure, as well as the miniaturization of mass spectrometers. Ambient mass spectrometry covers the former two, breaking through the restriction of traditional mass spectrometry in sample pretreatment and separation as well as the necessity of ionization at high vacuum, and determines trace substances in the complex matrix samples with high speed and high throughput. MPT mass spectrometry has been developed with many distinct characteristics in recent years, and initiated to the studies of PAHs in our laboratory. Based on these preliminary results, MPT mass spectrometry is uniquely suited to the direct study of PAHs and has the potential to provide insight to dynamic processes about Birch reduction, regular hydrogen addition as well as molecule robustness. Meanwhile, it will have a wide application prospect in PAHs detection, which points out a new development direction for the direct, fast detection of some isomers of PAHs in the future.

Furthermore, there is another outstanding mass spectrometry technique, mass spectrometry imaging [148,149,150,151,152], which can play a significant role in promoting the studies of PHAs. Compared with those common mass spectrometry techniques, mass spectrometry imaging can not only get the content information of PAHs throughout the whole sample, but also obtain the information about the distribution of PAHs on a media surface. These are exactly two-dimensional data, which will greatly facilitate the studies about the migration, aggregation and generation as well as evolution of PAHs in various media.

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
