# Peer review of "Contemporary Research Progress on the Detection of Polycyclic Aromatic Hydrocarbons"

_ijerph, 2022, doi:10.3390/ijerph19052790_

Round 1

Reviewer 1 Report

Authors present results of their review work on research progress on the detection of polycyclic aromatic hydrocarbons. The work is quite interesting but requires small changes prior to its acceptance. Especially English quality needs serious correction – it is very difficult to follow idea of Authors and reach aim of sentence You try to present.

Abstract is sufficiently informative; why are the keywords so “strangely” written with numbers?

Line 47: it is not result of “fusing”; please rephrase. They are not important in the universe… who says so? One can say that they are present or have been detected in many environmental compartments but it doesn’t mean that they are important.

58-61: listing diseases here is chaotic, please unify it.

63: impact of… is irreversible – but it is controversial statement if polycyclic hydrocarbons are omnipresent as You said before so human input into this “load” or “burden” must be discussed.

Line 73: again be careful – fuel emission is not source of PAHs, incomplete fuel burning is the source.

Line 79: hat does this sentence mean? “who did exposure”?

MPT is not sufficiently described in intro.

Line 112: give ref confirming occurrence of synergy here.

2.1. is very superficial; please extend and add references to confirm Your statement. Also in 2.3.

175: GC and HPLC are not detection methods – these are separation methods

3.1 – it is ultratrivial subchapter; what is more full of mistakes. So is 3.2. and 3.3. - these are ultrabasic information and promile of studies on PAHs. Authors barely scratch the surface of the problem not showing any significant details of them nor pros/cons of these methods.

Table 1 must go into electronic supplement or removed. It doesn’t present nothing ultraimprotant as is just cut off of many information for PAH vibrations.

Line 333: what do You mean by cross section?

Line 364: what is this “10-10”?

Line 380: this suggests that Min Zhang et al combined with SERS… Have Authors read their work prior to submission?

Line 417: linearity range didn’t start with 0; and correct the unit.

Line 452: relatively longer

Line 471: it really is simple state.

Line 494: MIT is here not a detection method, it is used to entrap analytes of interest to be subsequently extracted and determined with other methods.

4th chapter is most interesting (although full of English mistakes again) and should be core of this review. Remove fig 1 – it is too simple and selective to present it here in the review.

5th chapter must be extended by giving information on future development trends in this area.

Author Response

  1. Abstract is sufficiently informative; why are the keywords so “strangely” written with numbers?

Þ       This is due to our use of the magazine template in which the keywords are marked with numbers. This strangeness can be removed in post editing if necessary.  By the way, due to one of the author's affiliations changing, the forth organization State Key Laboratory of Organic Geochemistry and Guangdong-Hong Kong-Macao Joint Laboratory for Environmental Pollution and Control, Guangzhou Institute of Geochemistry, Chinese Academy of Sciences, Guangzhou 510640, China; [email protected].” has been changed into “National and Provincial Union Engineering Research Center for the Veterinary Herbal Medicine Resources and Initiative, Hunan Agricultural University, Changsha 410128, China; [email protected]” in the main text. (Page 1)

  1. Line 47: it is not result of “fusing”; please rephrase. They are not important in the universe… who says so? One can say that they are present or have been detected in many environmental compartments but it doesn’t mean that they are important.

Þ       Here, we rephrase the word “fusing” into condensing”. (Page2)Þ       About the question of whether PAHs is important in the universe, we have to admit our knowledge about this aspect is limit. We only refer to a recent science paper (reference 5). So the sentence “and are also important in the universe and interstellar medium, is changed into and appear in the universe and interstellar medium, (Page 2)

  1. 58-61: listing diseases here is chaotic, please unify it.

Þ       Sorry for these confusion. Now the sentences “According to epidemiological studies, workers exposed to PAHs for a long time are prone to cancer, especially skin cancer, leukemia, bladder cancer and so on, as well as nasopharyngeal carcinoma, gastric cancer and lung cancer. ” were changed into “ According to epidemiological studies, workers exposed to PAHs for a long time are prone to cancer, especially skin cancer, blood cancer, bladder cancer, nasopharyngeal cancer, gastric cancer as well as lung cancer.[10] (Page 2)

  1. 63: impact of… is irreversible – but it is controversial statement if polycyclic hydrocarbons are omnipresent as You said before so human input into this “load” or “burden” must be discussed.

Þ       We do agree the comment from the reviewer. This problem arise mainly from our poor expression and our relative limit knowledge about PAHs. Now we reconstructed this sentence in the form of “Because PAHs are difficult to metabolize and degrade, they have a significant impact on human health and ecological environment(Page 2)

  1. Line 73: again be careful – fuel emission is not source of PAHs, incomplete fuel burning is the source.

Þ       Following the suggestion of the review, the sentence “ So as the main source of PAHs, anthropogenic sources mainly include motor vehicle fuel emissions and coal-burning emissions,……” is changed into “So as the main source of PAHs, anthropogenic sources mainly include the incomplete combustion of motor vehicle fuel emissions and coal-burning emissions,……(Page 2)

  1. Line 79: hat does this sentence mean? “who did exposure”?

Þ       Sorry for our poor English expression. The sentence “…..in non-allergic children in the early stages of pyrene exposure ” is changed into “ …..in non-allergic children in the early stages of exposure to pyrene (Page 3)

  1. MPT is not sufficiently described in intro.

Þ       Really, we only mentioned slightly MPT-MS in the introduction owing to the construction of this article. Now some sentences about the description of MPT were supplied following the sentence“...as the far out novel results in our lab by using microwave plasma torch (MPT) mass spectrometry, and exhibited the versatility of MPT MS as a platform in the researches of PAHs. MPT ion source is a novel ambient ion generator and have multiple advantages, for example, simple construction of the device and easy operation, low power dissipation, relative high sensitivity, and suitable for many types of samples including metal elements and organic samples. [22-26] (Page 3)

  1. Line 112: give ref confirming occurrence of synergy here.

Þ       Sorry for making this big mistake, mainly due to the author’s negligence and imprudence. There is indeed no direct evidence confirming occurrence of synergy effect between the distributions of PAHs in atmosphere, soil and water, although one of the authors believe so. Thus, the original sentence “In general, the contents of PAHs in atmosphere, water and soil contend, restrict, and influence each other, exhibiting distinct synergistic effect.” was deleted.

  1. 2.1. is very superficial; please extend and add references to confirm Your statement. Also in 2.3.

Þ       Really, the author's expertise lies outside this area. The distribution and migration of PAHs in various medias are certainly important and specialized. Here we do our best to expand these two subsections. However, they may be removed if need.Þ       2.1 section As PAHs is a semi-volatile pollutant, the atmosphere is the main receptor and transmission and diffusion channel of PAHs emissions, as well as an important reservoir of PAHs. Atmosphere is the most important environmental medium in the migration and transformation of PAHs, and is also the most important medium of human exposure to PAHs pollution [29-31]. PAHs in atmospheric mainly come from pollutant emissions of industrial areas such as coking plants and incomplete combustion of civil coal-fired boilers [32, 33]. Yuan-ju Li et al. [34], recently discovered that a large amount of PAHs is also produced during catering processing. Clarifying the emission characteristics of PAHs and the distribution characteristics of particles can provide an important basis for analyzing the generation mechanism and source of atmospheric secondary organic aerosols (SOA).PAHs in soil and water also enter the atmosphere through evaporation. Some PAHs in the atmosphere will degrade under ultraviolet light, and also easily generate phenolic compounds, which will further react with NOx to generate nitrifies. Phenols and nitrifiers are one of the main components of atmospheric SOA, which will destroy the normal reaction cycle of NOx and O3, leading to the further increase of atmospheric O3 concen-tration. Therefore, PAHs is one of the reasons leading to serious atmospheric secondary pollution and high ozone concentration. [34] Atmospheric PAHs can also enter soil and water through descending migration. These processes are affected by their particle size, physical and chemical shapes, meteorological conditions and soil composition. The dis-tribution of PAHs is different in different regions. Levels of PAHs pollution are generally higher in inland areas than in coastal areas because daily human production and living activities are closer to inland areas. With the change of different seasons, the concentration level of PAHs in the environment also varies to a certain extent [7], showing a general trend of high concentration in winter and low concentration in summer. In floating dust, inhalable particles adhere to most PAHs, thus causing hidden dangers to the health development of humans and animals [8].”  (Page 4)Þ       2.3 section Soil is another important carrier of PAHs. Among the sources of PAHs in soil, besides natural sources, many human activities and industrial production take up most, including incomplete combustion of industrial fuels, atmospheric sedimentation and industrial sewage. Earlier researches [36] have shown that atmospheric deposition of PAHs is the most important in this part, probably accounting for more than 90%. The content of PAHs in soil has noteworthy regional and seasonal differences as that in air and water, which deeply indicates that PAHs migrates and converts among air, water and soil, forming an ecological closed loop. Multi-media migration is one of the important characteristics of semi-volatile organic pollutants such as PAHs, which also brings some difficulties to the researches of PAHs. It is necessary to developing a holistic study and comprehensive analysis of various data as well as simulating their migration and transformation behavior in all environmental media at the same time. The development of these new compatible methods and technologies relies more on modern analysis techniques and statistical processing methods from big data analysis.PAHs enter soil, causing damage to soil environment and normal working perfor-mance. PAHs in soil also have much more complicated environmental behaviors in soil, including adsorption, degradation and migration. Each process involves the influence of physical, chemical or biological environmental processes and is extremely complicated. For example, the adsorption of PAHs in soil affected by the soil surface chemical force on PAHs, as well as electrostatic force and van der Waals attraction from relatively long distance. Therefore, PAHs adsorption in soil has two different stages[37-39], naming fast process and slow process. The fast process leads to the adsorption of PAHs on the soil hydrophobic surface, while the slow process involves the migration of PAHs to the deep and inaccessible part of the soil matrix, which is easy to be absorbed and enriched by some vegetation[40, 41], thus affecting the agricultural safety production to a certain extent. Different adsorption extent depends strongly on the physical and chemical properties of PAHs and soil [42], such as polycyclic aromatic hydrocarbons, water solubility, soil particle size, soil organic carbon content, pH and temperature. Adsorption of PAHs in soil in turn affect their further actions in the environment, such as volatile, photolysis, hydrolysis, and the important factors in the process of biological degradation and so on. PAHs in soil are then transmitted through the food chain, eventually causing irreversible damage to humans and animals [43]. (Page 4)

  1. 175: GC and HPLC are not detection methods – these are separation methods

Þ       Sorry for wrong expression. Now the original sentence “Among them, HPLC and GC are the most common means of detection technology is mature and generally chose as the standard methods, but there are still insufficient.” is changed in to “ Among them, HPLC and GC are the most common means of analytical methods is mature and generally chose as the standard methods, but there are still insufficient. (Page 5)

  1. 3.1 – it is ultratrivial subchapter; what is more full of mistakes. So is 3.2. and 3.3. - these are ultrabasic information and promile of studies on PAHs. Authors barely scratch the surface of the problem not showing any significant details of them nor pros/cons of these methods.

Þ       We do agree the comments. Now, we supplied some comments about these analytical methods.For 3.1 section, a new paragraph was supplied as some comment as the end of this section. Although HPLC is almost the most commonly used analytical method, especially hybridized with other various detection approaches, it still seems a little inadequate. First, the detection ability of HPLC for the separated components will strongly depend on the detector. The detector has high sensitivity and low detection limit, which plays an important role in the detection and analysis of PAHs. How to develop suitable high-efficiency detector will be a promising approach. Second, high-powered separation device is need to be further developed. Commonly, HPLC combined with solid phase membrane extraction, which has some disadvantages such as complicated pretreatment and expensive instruments and reagents. More importantly, the current hybrid HPLC methods only give the information about content, ingredient. Extracting the much deeply information about the conversion and transition of each component of PAHs will be in-teresting and potential for the in-depth study involving PAHs. (Page 6)For 3.2 section, the paragraph “ Carlos Manzano et al. [40] developed a way to improve the separation of complex PAH mixtures (including 97 different parent, alkyl-, nitro-, oxy-, thio-, chloro-, bromo-, and high molecular weight PAHs) also using a two-dimensional GC-MS, that is GC×GC/TOF-MS by maximizing the orthogonality of different GC column combinations and improving the separation of PAHs from the sample matrix interferences, including unresolved complex mixtures. They used four different combinations of non-polar, polar, liquid crystal, and nano-stationary phase columns, each column combination can be optimized. They evaluated also every column for orthogonality using a method based on conditional entropy that considers the quantitative peak distribution in the entire 2D space (for completed chromatography data). Based the approach, they analyzed the atmospheric particulate sample of matter PM 2.5 from Beijing, China, and a soil sample and a sediment sample from overseas for complex mixtures of PAHs. The found that the highest chromatographic resolution, lowest synentropy, highest orthogonality, and lowest interference from UCM were achieved using a 10 m×0.15 mm×0.10μm LC-50 liquid crystal column in the first dimension and a 1.2m×0.10 mm×0.10μm NSP-35 nano-stationary phase column in the second dimension. They demonstrated that the use of this column combination in GC×GC/TOF-MS resulted in significantly shorter analysis times (176 min) for complex PAH mixtures compared to 1D GC/MS (257 min), as well as potentially reduced sample preparation time.” was supplied a comment as the end of this paragraph “ While, their results merely contained the contents of PAHs, without further exploring the correlations between each component of PAHs, in other words, the 2D-dimensional data is still underexplored. (Page 7)For 3.3 section, some little comments were supplied: However, for the sensitivity in this field, there is still plenty of room for improvement, especially when uniting un/vis spectograph, mainly owing to the tiny capillary diameter making light path too short. Maybe this can be overcome by optimal the light path such as employing multi-pass and so on. The biggest deficiency for CE may be the reproducibility is poor due to the electroosmotic flow changing with the composition of the sample. This point needs still a lot efforts.” was added to the end of the first paragraph in this section. (Page 8) Importantly, the authors claimed that using umbelliferone as an internal standard with appropriate electrolyte and sample compositions, rinse sequences and sample vial material they improved significantly the repeatability.” was added following “ Ludivine Ferey et al.[51] optimized cyclodextrin modified capillary electrophoresis with laser-induced fluorescence detection. The utilizing of a dual CD system, involving a mixture of one neutral CD and one anionic CD, enabled them to reach unique selectivity and achieved the best separation effect of 19 PAHs with efficiencies superior to 1.5×105 in 15 minutes for the first time. And the detection of PAHs in edible oil and real vegetable oil can be carried out by using internal standard with low LOQs in mg·L-1. (Page 8)

  1. Table 1 must go into electronic supplement or removed. It doesn’t present nothing ultraimprotant as is just cut off of many information for PAH vibrations.

Þ       In this new manuscript, the table 1 has been removed into the supplement and remarked as Table S1.

  1. Line 333: what do You mean by cross section?

Þ       Here, the phrase cross section represents the intersecting surface of the involving capillary tube. In general, the bigger it is , the more CE can withstand high voltages. For more clarity, the original “its cross sections” has been changed into “ the cross section of capillary tube (Page 7)

  1. Line 364: what is this “10-10”?

Þ       Sorry for this negligence. It should be 10-10. (Page 8)Þ       Analogously, “1.5×105” should be “ 1.5×105 (Page 8)

  1. Line 380: this suggests that Min Zhang et al combined with SERS… Have Authors read their work prior to submission?

Þ       We have read that paper sketchily. Maybe due to our poor English expressing, our article diverged the original statement. After checking carefully again the abstract, this original paragraph is changed into “ Adopting a hyphenated technique combining SERS with surface microextraction by using methanol and 1-propanethiol-modified silver nanoparticles, Min Zhang et al. [69] realized the in-situ on site analysis of PAHs on food contact materials. Wherein, the characteristic vibration of the C–C bond of parathyroid hormone at 1030 cm−1 was used as the internal standard for quantitative determination in this method, giving high uniform 1-propanethiol-modified silver nanoparticles. In the concentration range of 0.789-158 ng·cm2, the standardized SERS intensity showed a good linear relationship with fluoranthene concentration, the LOD was 0.27 ng·cm2. This detection method can realize the rapid screening of PAHs mixtures, making the analysis efficiency greatly improved.” 

  1. Line 417: linearity range didn’t start with 0; and correct the unit.

Þ       The data cited here is completely from ref 61(in original manuscript). We read carefully this reference again and found that the low limit of the linear range can be set to 0.1ng/ml. Thus, the original “The linear response of this method was in the range of 0-1000ng·ml-1(R≥0. 9988), the relative standard deviations (RSD) were in the range of 1.06% -1.67% (n=6). The LODs were in the range of 0. 072- 3.9 ng·ml-1.” Was changed into “ The linear response of this method was in the range of 0.1-1000ng·ml-1(R≥0. 9988), the relative standard deviations (RSD) were in the range of 1.06% -1.67% (n=6). The LODs were in the range of 0. 072- 3.9 ng·ml-1.(Page 9)

  1. Line 452: relatively longer

Þ       rather longer was changed into relatively longer (Page 10)

  1. Line 471: it really is simple state.

Þ       we give a simple state.” is changed into “ we only give a simple comment. (Page 10)

  1. Line 494: MIT is here not a detection method, it is used to entrap analytes of interest to be subsequently extracted and determined with other methods.

Þ       Surely, MIT is a very important analytical method for PAHs, which entrap the aim analytes for the further detection. Here, we restate this part. “ Molecular Imprinting Technology (MIT)[93] is another important analytical method, which is used to entrap analytes of interest for the subsequent detection. Usually, it  uses molecularly imprinted polymers to simulate the interaction between enzyme-substrate or antibody-antigen to recognize specifically printed molecules. Thus, MIT has been also been used in the analysis of PAHs.(Page 11)

  1. 4th chapter is most interesting (although full of English mistakes again) and should be core of this review. Remove fig 1 – it is too simple and selective to present it here in the review.

Þ       Thank you for your comment. Surely, we really focused the mass spectrometry method, which is something we're familiar with. Following this suggestion, Figure 1 is removed and the subsequent figures are re-numbered.

  1. 5th chapter must be extended by giving information on future development trends in this area.

Þ       We have given some comments about the future development techniques in the end of introduction part, including micro-fluidics, chemometrics, machine learning. These techniques or methods have all the huge potential in complex matrix samples, also can be applied in the detection of PHAs. In addition, we also look forward into the future in which GCXGC-MS and mass spectrometry imaging is applied in the studying of PHAs. Mass spectrometry imaging has the remarked merit with which the one can obtained the information about the distribution of PAHs on a media surface.So , the whole section now is changed into In this short article, we review the contemporary analytical methods for PAHs. The detection of PAHs is undoubtedly important in modern social life and scientifically re-searches. There are also many ways to detect PAHs. Unfortunately, some of them are flawed in traditional analysis methods. For example, high performance liquid chroma-tography pretreatment is excessively complicated, and the detection sensitivity of capil-lary electrophoresis needs still to be further improved. There are some analytical methods, which have not yet ploy their full role. For example, GCXGC-MS, in which the data is extended one-dimensional, is more suitable for exploring the implicit complicated relationships between multiple factors such as synergistic competition or much higher order relations. However it does not exert a prominent effect and achieve a breakthrough case, although it has become an important equipment for field atmospheric monitoring[43, 141]. In addition, it is still worth noting that there are still many novel powerful analytical approaches, such as micro-fluidics [142, 143], chemometrics [144, 145], and machine learning [146, 147] and so on. These new approaches have been applied in numerous aspects of cotemporary leading-edge researches. These techniques or methods have all the huge potential in complex matrix samples, also can be applied in the detection of PHAs. Limited to the author's knowledge, these methods and technologies have been rarely used in the studies of PAHs, thus we do not mention them regrettably. However, it is certain that they will shine surely in the research of PAHs in the near future.Completed realizing real-time and online detecting and analyzing at least comprises direct sample injection, and ionization technique under atmospheric pressure as well as the miniature of mass spectrometer. Ambient mass spectrometry covers the former two, breaking through the restriction of traditional mass spectrometry in sample pretreatment and separation as well as the necessary of ionization in high vacuum, and determines trace substances in the complex matrix samples with high speed and high throughput. MPT mass spectrometry has been developed with many distinct characters recent years and begun to the studies of PAHs in our laboratory. Based on these preliminary results, MPT mass spectrometry is uniquely suited the direct study of PAHs and has the potential to provide insight to dynamic processes about Birch reduction, regular hydrogen addition as well as molecule robustness. Meanwhile, it will have a wide application prospect in PAHs detection, which points out a new development direction for the direct, fast detection of some isomers of PAHs in the future. Again, there is another outstanding mass spectrometry technique, mass spectrometry imaging [148-152], can play a huge role in promoting the studies of PHAs. Relative to the common mass spectrometry which only get the content information of PAHs throughout the whole sample, mass spectrometry imaging can obtain obtained the information about the distribution of PAHs on a media surface. These are exactly a two-dimension data, which will greatly facilitate the studies about the migration, aggregation and generation as well as evolution of PAHs in various media. (Page 18)

Reviewer 2 Report

In this short review the authors focused on the contemporary analytical methods about Polycyclic aromatic hydrocarbons (PAHs). They started with a
brief review on the hazards, migration, distribution and traditional analysis methods of PAHs in recent years. They also presented the applications of the
mass spectrometry, especially microwave plasma torch mass spectrometry, in the detection of PAHs and their lab results by using microwave plasma torch mass spectrometry.

I have some questions and suggestions:
1.- Although many books and review articles are available on different aspects of PAHs chemistry, a modern description of the current state of analytical methods on PAHs is highly desirable. This review seems to meet this need. However, the manuscript does not allow to identify the importance of having a review article on the subject, the work seems to be a data collection.
2.- How many reviews have been published about it?. In this manuscript there are only 4 references to works published in 2021. The period covered by the review is not indicated.
3.- Why is Table 1 important?. What is the relationship between vibrational frequency and HPLC?.

I think the authors need to do more work on the review!.

Author Response

  1. Although many books and review articles are available on different aspects of PAHs chemistry, a modern description of the current state of analytical methods on PAHs is highly desirable. This review seems to meet this need. However, the manuscript does not allow to identify the importance of having a review article on the subject, the work seems to be a data collection.

Þ       We do really admit the comment from the reviewer. However, in this short review, we mainly presented the applications of mass spectrometry method in the detection of PAHs, especially our new experimental result. Certainly, much deeper discussion will be presented in the coming publications. 

  1. How many reviews have been published about it? In this manuscript there are only 4 references to works published in 2021. The period covered by the review is not indicated.

Þ       PAHs is an old while new topic. The related researches covers many areas. However, the primary goal of this article related to the analytical methods and techniques for PHAs, essential in the category of analytical chemistry. In this field several aspects have been recently reviewed, such as, Somandla Ncube et al. Trends in Analytical Chemistry, 2018, 99, 101-116.; Elham Mansouri et al. 2020,1-9. The latter is about the ultraviolet-based methods, we have cited it and replenished a paragraph in this revised version. In 2020, Elham Mansouri et al. [84] reviewed critically and objectively ultraviolet-based methods especially in high-performance liquid chromatography-ultraviolet. They mainly discussed high-performance liquid chromatography ultraviolet-diode array detector and ultraviolet-fluorescent detector and their applications in scientific studies to measure polycyclic aromatic hydrocarbons concentration. The review also gives useful and comprehensive information about valuable methods for future researches on PAHs.(Page 10)Meanwhile, we mainly focused ambient mass spectrometry, especially the MPT mass spectrometry. Limited to the author’s knowledge, this aspect is rare. We also gave some outlooks on the new techniques which have potential applied in the studies of PHAs, and cited some latest works published in 2020or 2021, parts as shown in below. In this short article, we review the contemporary analytical methods for PAHs. The detection of PAHs is undoubtedly important in modern social life and scientifically re-searches. There are also many ways to detect PAHs. Unfortunately, some of them are flawed in traditional analysis methods. For example, high performance liquid chroma-tography pretreatment is excessively complicated, and the detection sensitivity of capil-lary electrophoresis needs still to be further improved. There are some analytical methods, which have not yet ploy their full role. For example, GCXGC-MS, in which the data is extended one-dimensional, is more suitable for exploring the implicit complicated relationships between multiple factors such as synergistic competition or much higher order relations. However it does not exert a prominent effect and achieve a breakthrough case, although it has become an important equipment for field atmospheric monitoring[43, 141]. In addition, it is still worth noting that there are still many novel powerful analytical approaches, such as micro-fluidics [142, 143], chemometrics [144, 145], and machine learning [146, 147] and so on. These new approaches have been applied in numerous aspects of cotemporary leading-edge researches. These techniques or methods have all the huge potential in complex matrix samples, also can be applied in the detection of PHAs. Limited to the author's knowledge, these methods and technologies have been rarely used in the studies of PAHs, thus we do not mention them regrettably. However, it is certain that they will shine surely in the research of PAHs in the near future.Completed realizing real-time and online detecting and analyzing at least comprises direct sample injection, and ionization technique under atmospheric pressure as well as the miniature of mass spectrometer. Ambient mass spectrometry covers the former two, breaking through the restriction of traditional mass spectrometry in sample pretreatment and separation as well as the necessary of ionization in high vacuum, and determines trace substances in the complex matrix samples with high speed and high throughput. MPT mass spectrometry has been developed with many distinct characters recent years and begun to the studies of PAHs in our laboratory. Based on these preliminary results, MPT mass spectrometry is uniquely suited the direct study of PAHs and has the potential to provide insight to dynamic processes about Birch reduction, regular hydrogen addition as well as molecule robustness. Meanwhile, it will have a wide application prospect in PAHs detection, which points out a new development direction for the direct, fast detection of some isomers of PAHs in the future. Again, there is another outstanding mass spectrometry technique, mass spectrometry imaging [148-152], can play a huge role in promoting the studies of PHAs. Relative to the common mass spectrometry which only get the content information of PAHs throughout the whole sample, mass spectrometry imaging can obtain obtained the information about the distribution of PAHs on a media surface. These are exactly a two-dimension data, which will greatly facilitate the studies about the migration, aggregation and generation as well as evolution of PAHs in various media. (Page 18)

  1. Why is Table 1 important? What is the relationship between vibrational frequency and HPLC? I think the authors need to do more work on the review!

Þ       It is not such important. In this revised manuscript, table 1 has been removed into the supplement following the suggestion from another reviewer.

Round 2

Reviewer 1 Report

Well, I am afraid Authors still didn’t do the homework especially on quality of English. I still cannot give my positive opinion on this manuscript. Authors in many situations underline their lack of knowledge or experience in this topic thus my question is: what made You feel “strong enough” in this area of expertise to prepare review work?

Work still has dozens of English mistakes which are again impossible to be enumerated.

What does it mean? In 205-213 that “More detecting techniques can refer to a review article”? and in 206 You write about GC and HPLC and then again You write about them “Among them, HPLC and  GC are the most common means of analytical methods is mature and generally chose as  the standard methods,”? – especially this part “common means of analytical methods is mature …”. And why are they not sufficient? If You refer to sample preparation then GC and LC have almost nothing to do with it.

Please rephrase “Because PAHs are difficult to metabolize and degrade, they have a significant impact on human health and ecological environment” to “Because PAHs are difficult to metabolize and degrade, they have a significant impact on human health and entire ecosystems”.

Line 232: why “-1” are not in upper cases? Such mistakes are present in many locations what justifies my statement that Authors didn’t pay too much attention to the reviewing process.

Lines 258-269: this paragraph is again chaotic; I do not understand any idea standing behind it. For instance what does it mean that “More importantly, the current hybrid HPLC methods only give the information about content, ingredient.”?

Line 317: these numerical values refer to selectivity, accuracy or low LOD? And for which of PAHs among studied ones?

Line 338: what does it mean that results merely contained the content?

Line 358: what is “5-ms”? milliseconds? Or columns of 5 m length?

Line 400: what is unit of this high efficiency?

Line 552: add space in front of ref.

Line 561: EU means for Authors the European Union or element Europium? Then “Eu”.

Line 664: there is a second set of peaks…

Lines 784-790: this must be English checked

Author Response

Well, I am afraid Authors still didn’t do the homework especially on quality of English. I still cannot give my positive opinion on this manuscript. Authors in many situations underline their lack of knowledge or experience in this topic thus my question is: what made You feel “strong enough” in this area of expertise to prepare review work?

Work still has dozens of English mistakes which are again impossible to be enumerated.

Þ      Indeed, we are all not experts in this field. Our knowledge about PAHs come completely from the recent literature collection. The purpose of our project is mainly developing the detection technique of ambient mass spectrometry for PAHs. In order to find the difference between our technique about MPT mass spectrometry and other conventional detection techniques, we have to survey all kinds of existing analytical methods for PAHs, and offer our own opinion, then it produces the theme of this review. Þ      Sorry for making so many expression mistakes. We have read carefully the manuscript this time, and also invited a rigorous English teacher and a PhD doctor from Warwick University read seriously the manuscripts and modify some expressions. All of modifications have been highlighted in this new version.

  1. What does it mean? In 205-213 that “More detecting techniques can refer to a review article”? and in 206 You write about GC and HPLC and then again You write about them “Among them, HPLC and GC are the most common means of analytical methods is mature and generally chose as  the standard methods,”? – especially this part “common means of analytical methods is mature …”. And why are they not sufficient? If You refer to sample preparation then GC and LC have almost nothing to do with it.

Þ      Sorry for unclear expression. This part has been rewritten in “ More detection techniques and related works can refer to a review article [44] and references therein. Among all the mature analytical methods, HPLC and GC are the most common ones and are generally chosen as the standard methods.(Page 5)

  1. Line 232: why “-1” are not in upper cases? Such mistakes are present in many locations what justifies my statement that Authors didn’t pay too much attention to the reviewing process.

Þ      I am so sorry for making such stupid mistakes. It is due to format conversion process, but mostly due to our negligence and carelessness. We have read carefully this manuscript and corrected these types of mistakes.

  1. Lines 258-269: this paragraph is again chaotic; I do not understand any idea standing behind it. For instance what does it mean that “More importantly, the current hybrid HPLC methods only give the information about content, ingredient.”?

Þ      Sorry for chaotic expression. Now this paragraph has been rewritten as following: “ Although HPLC is widely used in detecting PAHs, it has still some deficiency to be improved and perfected. First, the application of HPLC strongly depends on the detectors. The detector with high sensitivity, low detection limit and universality plays an important role in the detection and analysis of PAHs. Therefore, developing a suitable high-efficiency detector may be of certain significance. Second, high-powered separation device needs to be further developed. Currently, HPLC combined with solid phase membrane extraction has still some disadvantages to ameliorate, such as complicated pretreatment and expensive instruments and reagents. More importantly, the current hybrid HPLC methods can only show the information about the ingredients and content of various samples. However, the relevance and tangle of different ingredients are yet hidden in raw data. Exploiting deeply the information about the conversion, transition and synergy between different components of PAHs will be interesting and potential for the in-depth study involving PAHs." (Page 6)

  1. Line 317: these numerical values refer to selectivity, accuracy or low LOD? And for which of PAHs among studied ones?

Þ      These values in parenthesis denote the LOD range for 16 kinds of PAHs detected in their analytical method. Now, in this new version, “low LOD (0.22- 2.20 ng·L-1)” have been changed into “ low LOD values (0.22 ng·L-1 for Benzo[k]fluoranthene - 2.20 ng·L-1 for Phenanthrene, respectively) (Page 7)

  1. Line 338: what does it mean that results merely contained the content?

Þ      Well, this is really a new insight from the corresponding author, about the GC×GC data. It consisting of two dimensional data should provide more in-depth information about relevance and tangle of different ingredients, on which the author’s group is working. Now the original sentence has been changed into “ While, their results merely contained the information of ingredients and contents of PAHs, without further exploring the correlations and synergy between different components of PAHs, in other words, the 2D-dimensional data is still underexplored.(Page 7)

  1. Line 358: what is “5-ms”? milliseconds? Or columns of 5 m length?

Þ      After checking the original reference, we are sure that it should denote some kind of type specification. For clarity, the original sentence “ In their studies, the six methylchrysene isomers and the PAH compounds with a molecular weight of 302 Daltons in fat-containing foods attained a better chromatographic separation in comparison with a 5-ms column.” has been changed into “In their studies, the six methylchrysene isomers and the PAH compounds with a molecular weight of 302 Daltons in fat-containing foods attained a better chromatographic separation with VF-17ms GC column. (Page 7)7. Line 400: what is unit of this high efficiency?Þ      Here, the “high efficiency” mainly refer to higher plate number and higher separation efficiency. Now, it has been supplied in this new version, “high efficiency” is changed into “higher plate number and higher separation efficiency(Page 8)

  1. Line 552: add space in front of ref.

Þ      Thank you for this reminder. So we did and corrected other similar problems. (Page 11 and others with highlights)

  1. Line 561: EU means for Authors the European Union or element Europium? Then “Eu”.

Þ      Sorry for making such stupid mistakes. EU should be Eu, the element. (Page 11)

  1. Line 664: there is a second set of peaks…

Þ      Sorry for making such stupid mistakes. We have corrected this kind of mistake. (Page 13)

  1. Lines 784-790: this must be English checked

Þ      The last paragraph has been changed into “ Furthermore, there is another outstanding mass spectrometry technique, mass spectrometry imaging [148-152], which can play a significant role in promoting the studies of PHAs. Compared with those common mass spectrometry techniques, mass spectrometry imaging can not only get the content information of PAHs throughout the whole sample, but also obtain the information about the distribution of PAHs on a media surface. These are exactly two-dimension data, which will greatly facilitate the studies about the migration, aggregation and generation as well as evolution of PAHs in various media. (Page 19)

Reviewer 2 Report

In this short review the authors focused on the contemporary analytical methods about Polycyclic aromatic hydrocarbons (PAHs). They started with a brief review on the hazards, migration, distribution and traditional analysis methods of PAHs in recent years. They also presented the applications of the mass spectrometry, especially microwave plasma torch mass spectrometry, in the detection of PAHs and their lab results by using microwave plasma torch (MPT) mass spectrometry.

I had some questions for the original version of the manuscript:

1.- Although many books and review articles are available on different aspects of PAHs chemistry, a modern description of the current state of analytical methods on PAHs is highly desirable. This review seems to meet this need. However, the manuscript does not allow to identify the importance of having a review article on the subject, the work seems to be a data collection.

2.- How many reviews have been published about it?. In this manuscript there are only 4 references to works published in 2021. The period covered by the review is not indicated.

3.- Why is Table 1 important?. What is the relationship between vibrational frequency and HPLC?.

The new version of the manuscript includes answers to the questions. I think it is appropriate for the journal.

Author Response

thanks a lot